# Optimal Number of Choices in Rating Contexts

**Sam Ganzfried [1,*] and Farzana Beente Yusuf [2]**

[1]  Ganzfried Research, Miami Beach, FL 33139, USA
[2]  School of Computing and Information Sciences, Florida International University, Miami, FL 33199, USA; fyusu003@fiu.edu
*  Correspondence: sam@ganzfriedresearch.com

**Abstract:** In many settings, people must give numerical scores to entities from a small discrete set—for instance, rating physical attractiveness from 1–5 on dating sites, or papers from 1–10 for conference reviewing. We study the problem of understanding when using a different number of options is optimal. We consider the case when scores are uniform random and Gaussian. We study computationally when using 2, 3, 4, 5, and 10 options out of a total of 100 is optimal in these models (though our theoretical analysis is for a more general setting with $k$ choices from $n$ total options as well as a continuous underlying space). One may expect that using more options would always improve performance in this model, but we show that this is not necessarily the case, and that using fewer choices—even just two—can surprisingly be optimal in certain situations. While in theory for this setting it would be optimal to use all 100 options, in practice, this is prohibitive, and it is preferable to utilize a smaller number of options due to humans' limited computational resources. Our results could have many potential applications, as settings requiring entities to be ranked by humans are ubiquitous. There could also be applications to other fields such as signal or image processing where input values from a large set must be mapped to output values in a smaller set.

**Keywords:** recommender system; ranking; survey design; decision analysis; applied probability; quantization

## 1. Introduction

Humans rate items or entities in many important settings. For example, users of dating websites and mobile applications rate other users' physical attractiveness, teachers rate scholarly work of students, and reviewers rate the quality of academic conference submissions. In these settings, the users assign a numerical (integral) score to each item from a small discrete set. However, the number of options in this set can vary significantly between applications, and even within different instantiations of the same application. For instance, for rating attractiveness, three popular sites all use a different number of options. On "Hot or Not," users rate the attractiveness of photographs submitted voluntarily by other users on a scale of 1–10. These scores are aggregated and the average is assigned as the overall "score" for a photograph. On the dating website OkCupid, users rate other users on a scale of 1–5 (if a user rates another user 4 or 5, then the rated user receives a notification). In addition, on the mobile application Tinder, users "swipe right" (green heart) or "swipe left" (red X) to express interest in other users (two users are allowed to message each other if they mutually swipe right), which is essentially equivalent to using a binary $\{1, 2\}$ scale. Education is another important application area requiring human ratings. For the

2016 International Joint Conference on Artificial Intelligence, reviewers assigned a "Summary Rating" score from $-5$–$5$ (equivalent to 1–10) for each submitted paper. The papers are then discussed and scores aggregated to produce an acceptance or rejection decision based on the average of the scores.

Despite the importance and ubiquity of the problem, there has been little fundamental research done on the problem of determining the optimal number of options to allow in such settings. We study a model in which users have an underlying integral ground truth score for each item in $\{1, \ldots, n\}$ and are required to submit an integral rating in $\{1, \ldots, k\}$, for $k << n$. (For ease of presentation, we use the equivalent formulation $\{0, \ldots, n-1\}$, $\{0, \ldots, k-1\}$.) We use two generative models for the ground truth scores: a uniform random model in which the fraction of scores for each value from 0 to $n-1$ is chosen uniformly at random (by choosing a random value for each and then normalizing), and a model where scores are chosen according to a Gaussian distribution with a given mean and variance. We then compute a "compressed" score distribution by mapping each full score $s$ from $\{0, \ldots, n-1\}$ to $\{0, \ldots, k-1\}$ by applying

$$s \leftarrow \left\lfloor \frac{s}{\left(\frac{n}{k}\right)} \right\rfloor. \tag{1}$$

We then compute the average "compressed" score $a_k$, and compute its error $e_k$ according to

$$e_k = \left| a_f - \frac{n-1}{k-1} \cdot a_k \right|, \tag{2}$$

where $a_f$ is the ground truth average. The goal is to pick $\mathrm{argmin}_k e_k$ (in our simulations, we also consider a metric of the frequency at which each value of $k$ produces lowest error over all the items that are rated). While there are many possible generative models and cost functions, these seem to be the most natural, and we plan to study alternative choices in future work.

We derive a closed-form expression for $e_k$ that depends on only a small number ($k$) of parameters of the underlying distribution for an arbitrary distribution.This allows us to exactly characterize the performance of using each number of choices. In simulations, we repeatedly compute $e_k$ and compare the average values. We focus on $n = 100$ and $k = 2, 3, 4, 5, 10$, which we believe are the most natural and interesting choices for initial study.

One could argue that this model is somewhat "trivial" in the sense that it would be optimal to set $k = n$ to permit all the possible scores, as this would result in the "compressed" scores agreeing exactly with the full scores. However, there are several reasons that would lead us to prefer to select $k << n$ in practice (as all of the examples previously described have done), thus making this analysis worthwhile. It is much easier for a human to assign a score from a small set than from a large set, particularly when rating many items under time constraints. We could have included an additional term into the cost function $e_k$ that explicitly penalizes larger values of $k$, which would have a significant effect on the optimal value of $k$ (providing a favoritism for smaller values). However, the selection of this function would be somewhat arbitrary and would make the model more complex, and we leave this for future study. Given that we do not include such a penalty term, one may expect that increasing $k$ will always decrease $e_k$ in our setting. While the simulations show a clear negative relationship, we show that smaller values of $k$ actually lead to smaller $e_k$ surprisingly often. These smaller values would receive further preference with a penalty term.

One line of related theoretical research that also has applications to the education domain studies the impact of using finely grained numerical grades (100, 99, 98) vs. coarse letter grades (A, B, C) [1]. They conclude that, if students care primarily about their rank relative to the other students, they are often best motivated to work by assigning them coarse categories than exact numerical scores. In a setting of "disparate" student abilities, they show that the optimal absolute grading scheme is always coarse. Their model is game-theoretic; each player (student) selects an effort level, seeking to optimize a utility

function that depends on both the relative score and effort level. Their setting is quite different from ours in many ways. For one, they study a setting where it is assumed that the underlying "ground truth" score is known, yet may be disguised for strategic reasons. In our setting, the goal is to approximate the ground truth score as closely as possible.

While we are not aware of prior theoretical study of our exact problem, there have been experimental studies on the optimal number of options on a "Likert scale" [2–6]. The general conclusion is that "the optimal number of scale categories is content specific and a function of the conditions of measurement." [7] There has been study of whether including a "mid-point" option (i.e., the middle choice from an odd number) is beneficial. One experiment demonstrated that the use of the mid-point category decreases as the number of choices increases: 20% of respondents choose the mid-point for 3 and 5 options while only 7% did for $7, 9, \ldots, 19$ [8]. They conclude that it is preferable to either not include a mid-point at all or use a large number of options. Subsequent experiments demonstrated that eliminating a mid-point can reduce social desirability bias, which results from respondents' desires to please the interviewer or not give a perceived socially unacceptable answer [7]. There has also been significant research on questionnaire design and the concept of "feeling thermometers," particularly from the fields of psychology and sociology [9–14]. One study concludes from experimental data: "in the measurement of satisfaction with various domains of life, 11-point scales clearly are more reliable than comparable 7-point scales" [15]. Another study shows that "people are more likely to purchase gourmet jams or chocolates or to undertake optional class essay assignments when offered a limited array of six choices rather than a more extensive array of 24 or 30 choices" [16]. Since the experimental conclusions are dependent on the specific datasets and seem to vary from domain to domain, we choose to focus on formulating theoretical models and computational simulations, though we also include results and discussion from several datasets.

We note that we are not necessarily claiming that our model or analysis perfectly models reality or the psychological phenomena behind how humans actually behave. We are simply proposing simple and natural models that, to the best of our knowledge, have not been studied before. The simulation results seem somewhat counterintuitive and merit study on their own. We admit that further study is needed to determine how realistic our assumptions are for modeling human behavior. For example, some psychology research suggests that human users may not actually have an underlying integral ground truth value [17]. Research from the recommender systems community indicates that while using a coarser granularity for rating scales provides less absolute predictive value to users, it can be viewed a providing more value if viewed from an alternative perspective of preference bits per second [18].

Some work considers the setting where ratings over $\{1, \ldots, 5\}$ are mapped into a binary "thumbs up"/"thumbs down" (analogously to the swipe right/left example for Tinder above) [19]. Generally, users mapped original ratings of 1 and 2 to "thumbs down" and original ratings of 3, 4, and 5 to "thumbs up," which can be viewed as being similar to the floor compression procedure described above. We consider a more generalized setting where ratings over $\{1, \ldots, n\}$ are mapped down to a smaller space (which could be binary but may have more options). In addition, we also consider a rounding compression technique in addition to the flooring compression.

Some prior work has presented an approach for mapping continuous prediction scores to ordinal preferences with heterogeneous thresholds that is also applicable to mapping continuous-valued 'true preference' scores [20]. We note that our setting can apply straightforwardly to provide continuous-to-ordinal mapping in the same way as it performs ordinal-to-ordinal mapping initially (In fact, for our theoretical analysis and for the Jester dataset, we study our mapping is continuous-to-ordinal). An alternative model assumes that users compare items with pairwise comparisons which form a weak ordering, meaning that some items are given the same "mental rating," while, for our setting, the ratings would be much more likely to be unique in the fine-grained space of ground-truth scores [21,22]. Note that there has also been exploration within the data mining and AI communities for determining the optimal

of clusters to use for unsupervised learning algorithms [23–25]. In comparison to prior work, the main takeaway from our work is the closed-form expression for simple natural models, and the new simulation results that show precisely for the first time how often each number of choices is optimal using several metrics (number of times it produces lowest error and the lowest average error). We include experiments on datasets from several domains for completeness, though, as prior work, it has shown that results can vary significantly between datasets, and further research from psychology and social science is needed to make more accurate predictions of how humans actually behave in practice. We note that our results could also have impact outside of human user systems—for example, to the problems of "quantization" and data compression in signal processing.

## 2. Theoretical Characterization

Suppose scores are given by continuous probability density function (pdf) $f$ (with cumulative distribution function (cdf) $F$) on $(0, 100)$, and we wish to compress them to two options, $\{0, 1\}$. Scores below 50 are mapped to 0, and above 50 to 1. The average of the full distribution is $a_f = E[X] = \int_{x=0}^{100} x f(x) dx$. The average of the compressed version is

$$a_2 = \int_{x=0}^{50} 0 f(x) dx + \int_{x=50}^{100} 1 f(x) dx = 1 - F(50).$$

Thus, $e_2 = |a_f - 100(1 - F(50))| = |E[X] - 100 + 100 F(50)|$. For three options,

$$
\begin{aligned}
a_3 &= \int_{x=0}^{\frac{100}{3}} 0 f(x) dx + \int_{x=\frac{100}{3}}^{\frac{200}{3}} 1 f(x) dx + \int_{x=\frac{200}{3}}^{100} 2 f(x) dx \\
&= 2 - F(100/3) - F(200/3), \\
e_3 &= |a_f - 50(2 - F(100/3) - F(200/3))| \\
&= |E[X] - 100 + 50 F(100/3) + 50 F(200/3)|.
\end{aligned}
$$

In general, for $n$ total and $k$ compressed options,

$$
\begin{aligned}
a_k &= \sum_{i=0}^{k-1} \int_{x=\frac{ni}{k}}^{\frac{n(i+1)}{k}} i f(x) dx \\
&= \sum_{i=0}^{k-1} \left[ i \left( F\left( \frac{n(i+1)}{k} \right) - F\left( \frac{ni}{k} \right) \right) \right] \\
&= (k-1) F(n) - \sum_{i=1}^{k-1} F\left( \frac{ni}{k} \right) \\
&= (k-1) - \sum_{i=1}^{k-1} F\left( \frac{ni}{k} \right),
\end{aligned}
$$

$$
\begin{aligned}
e_k &= \left| a_f - \frac{n}{k-1} \left( (k-1) - \sum_{i=1}^{k-1} F\left( \frac{ni}{k} \right) \right) \right| \\
&= \left| E[X] - n + \frac{n}{k-1} \sum_{i=1}^{k-1} F\left( \frac{ni}{k} \right) \right|.
\end{aligned}
\tag{3}
$$

Equation (3) allows us to characterize the relative performance of choices of $k$ for a given distribution $f$. For each $k$, it requires only knowing $k$ statistics of $f$ (the $k - 1$ values of $F\left(\frac{ni}{k}\right)$ plus $E[X]$). In practice, these could likely be closely approximated from historical data for small $k$ values (though prior work has pointed out that there may be some challenges in order to closely approximate the cdf values of the ratings from historical data, due to the historical data not being sampled at random from the true rating distribution [26]).

As an example, we see that $e_2 < e_3$ iff

$$\left|E[X] - 100 + 100F(50)\right| < \left|E[X] - 100 + 50F\left(\frac{100}{3}\right) + 50F\left(\frac{200}{3}\right)\right|.$$

Consider a full distribution that has half its mass right around 30 and half its mass right around 60 (Figure 1). Then, $a_f = E[X] = 0.5 \times 30 + 0.5 \times 60 = 45$. If we use $k = 2$, then the mass at 30 will be mapped down to 0 (since $30 < 50$) and the mass at 60 will be mapped up to 1 (since $60 > 50$) (Figure 2). Thus, $a_2 = 0.5 \times 0 + 0.5 \times 1 = 0.5$. Using normalization of $\frac{n}{k} = 100$, $e_2 = |45 - 100(0.5)| = |45 - 50| = 5$. If we use $k = 3$, then the mass at 30 will also be mapped down to 0 (since $0 < \frac{100}{3}$), but the mass at 60 will be mapped to 1 (not the maximum possible value of 2 in this case), since $\frac{100}{3} < 60 < \frac{200}{3}$ (Figure 2). Thus, again $a_3 = 0.5 \times 0 + 0.5 \times 1 = 0.5$, but now, using normalization of $\frac{n}{k} = 50$, we have $e_3 = |45 - 50(0.5)| = |45 - 25| = 20$. Thus, surprisingly, in this example, allowing more ranking choices actually significantly increases error.

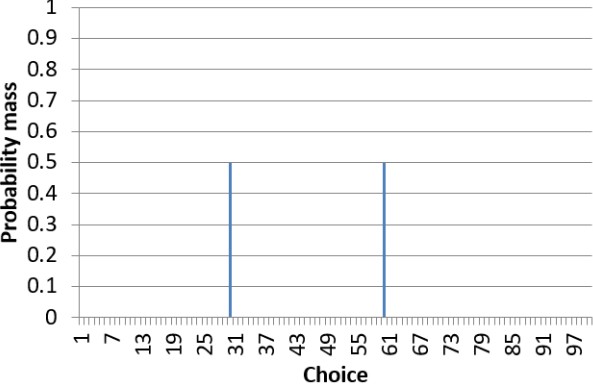

**Figure 1.** Example distribution for which compressing with $k = 2$ produces lower error than $k = 3$.

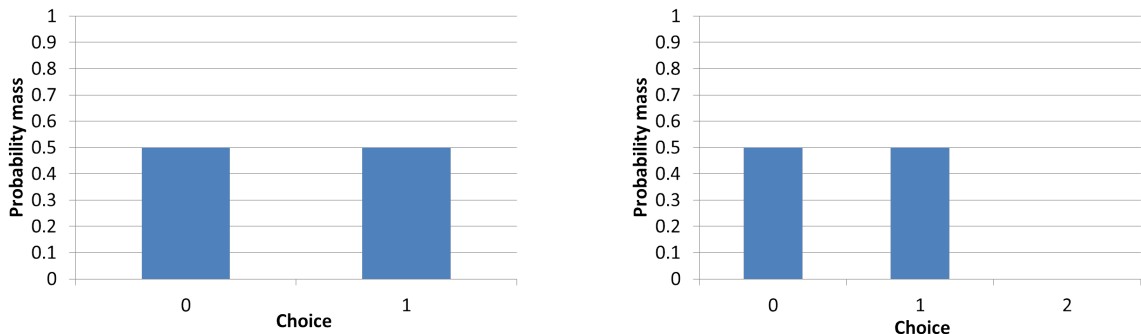

**Figure 2.** Compressed distributions using $k = 2$ and $k = 3$ for example from Figure 1.

If we happened to be in the case where both $a_2 \leq a_f$ and $a_3 \leq a_f$, then we could remove the absolute values and reduce the expression to see that $e_2 < e_3$ iff $\int_{x=\frac{100}{3}}^{50} f(x)dx < \int_{x=50}^{\frac{200}{3}} f(x)dx$. Note that the conditions $a_2 \leq a_f$ and $a_3 \leq a_f$ correspond to the constraints that $E[X] \geq 100(1 - F(50))$, $E[X] \geq 50(2 - F\left(\frac{100}{3}\right) - F\left(\frac{200}{3}\right))$. Taking these together, one specific set of conditions for which $e_2 < e_3$ is if both of the following are met:

$$\int_{x=\frac{100}{3}}^{50} f(x)dx < \int_{x=50}^{\frac{200}{3}} f(x)dx$$

$$E[X] \geq \max\left\{100\left(1 - F(50)\right), 50\left(2 - F\left(\frac{100}{3}\right) - F\left(\frac{200}{3}\right)\right)\right\}.$$

We can next consider the case where both $a_2 \geq a_f$ and $a_3 \geq a_f$. Here, we can remove the absolute values and switch the direction of the inequality to see that $e_2 < e_3$ iff $\int_{x=\frac{100}{3}}^{50} f(x)dx > \int_{x=50}^{\frac{200}{3}} f(x)dx$. Note that the conditions $a_2 \geq a_f$ and $a_3 \geq a_f$ correspond to the constraints that $E[X] \leq 100(1 - F(50))$, $E[X] \leq 50(2 - F\left(\frac{100}{3}\right) - F\left(\frac{200}{3}\right))$. Taking these together, a second set of conditions for which $e_2 < e_3$ is if both of the following are met: $\int_{x=\frac{100}{3}}^{50} f(x)dx > \int_{x=50}^{\frac{200}{3}} f(x)dx$, $E[X] \leq \min\left\{100\left(1 - F(50)\right), 50\left(2 - F\left(\frac{100}{3}\right) - F\left(\frac{200}{3}\right)\right)\right\}$.

For the case $a_2 \leq a_f \leq a_3$, the conditions are that

$$E[X] < 100 - 100\int_{x=0}^{\frac{100}{3}} f(x)dx - 75\int_{x=\frac{100}{3}}^{50} f(x)dx - 50\int_{x=50}^{\frac{100}{3}} f(x)dx$$

$$100\left(1 - F(50)\right) \leq E[X] \leq 50\left(2 - F\left(\frac{100}{3}\right) - F\left(\frac{200}{3}\right)\right).$$

In addition, finally, for $a_3 \leq a_f \leq a_2$,

$$E[X] > 100 - 100\int_{x=0}^{\frac{100}{3}} f(x)dx - 75\int_{x=\frac{100}{3}}^{50} f(x)dx - 50\int_{x=50}^{\frac{100}{3}} f(x)dx$$

$$100\left(1 - F(50)\right) \geq E[X] \geq 50\left(2 - F\left(\frac{100}{3}\right) - F\left(\frac{200}{3}\right)\right).$$

Using $k = 2$ outperforms $k = 3$ ($n = 100$) if and only if one of these four sets of conditions holds.

## 3. Rounding Compression

An alternative model we could have considered is to use rounding to produce the compressed scores as opposed to using the floor function from Equation (1). For instance, for the case $n = 100, k = 2$, instead of dividing $s$ by 50 and taking the floor, we could instead partition the points according to whether they are closest to $t_1 = 25$ or $t_2 = 75$. In the example above, the mass at 30 would be mapped to $t_1$ and the mass at 60 would be mapped to $t_2$. This would produce a compressed average score of $a_2 = \frac{1}{2} \times 25 + \frac{1}{2} \times 75 = 50$. No normalization would be necessary, and this would produce an error of $e_2 = |a_f - a_2| = |45 - 50| = 5$, as the floor approach did as well. Similarly, for $k = 3$, the region midpoints will be $q_1 = \frac{100}{6}$, $q_2 = 50$, $q_3 = \frac{500}{6}$. The mass at 30 will be mapped to $q_1 = \frac{100}{6}$ and the mass at 60 will be mapped to $q_2 = 50$. This produces a compressed average score of $a_3 = \frac{1}{2} \times \frac{100}{6} + \frac{1}{2} \times 50 = \frac{100}{3}$. This produces an error of $e_3 = |a_f - a_3| = \left|45 - \frac{100}{3}\right| = \frac{35}{3} = 11.67$. Although the error for $k = 3$ is smaller than for the floor case,

it is still significantly larger than $k = 2$'s, and using two options still outperforms using three for the example in this new model.

In general, this approach would create $k$ "midpoints" $\{m_i^k\}$: $m_i^k = \frac{n(2i-1)}{2k}$. For $k = 2$, we have

$$
\begin{aligned}
a_2 &= \int_{x=0}^{50} 25 + \int_{x=50}^{100} 75 = 75 - 50F(50), \\
e_2 &= |a_f - (75 - 50F(50))| = |E[X] - 75 + 50F(50)|.
\end{aligned}
$$

One might wonder whether the floor approach would ever outperform the rounding approach (in the example above, the rounding approach produced lower error $k = 3$ and the same error for $k = 2$). As a simple example to see that it can, consider the distribution with all mass on 0. The floor approach would produce $a_2 = 0$ giving an error of 0, while the rounding approach would produce $a_2 = 25$ giving an error of 25. Thus, the superiority of the approach is dependent on the distribution. We explore this further in the experiments.

For three options,

$$
\begin{aligned}
a_3 &= \int_0^{\frac{100}{3}} \frac{100}{6} f(x) + \int_{\frac{100}{3}}^{\frac{200}{3}} 50f(x) + \int_{\frac{200}{3}}^{100} \frac{500}{6} f(x) \\
&= \frac{500}{6} - \frac{100}{3} F\left(\frac{100}{3}\right) - \frac{100}{3} F\left(\frac{200}{3}\right), \\
e_3 &= \left| E[X] - \frac{500}{6} + \frac{100}{3} F\left(\frac{100}{3}\right) + \frac{100}{3} F\left(\frac{200}{3}\right) \right|.
\end{aligned}
$$

For general $n$ and $k$, analysis as above yields

$$
\begin{aligned}
a_k &= \sum_{i=0}^{k-1} \int_{x=\frac{ni}{k}}^{\frac{n(i+1)}{k}} m_{i+1}^k f(x)dx = \frac{n(2k-1)}{2k} - \frac{n}{k} \sum_{i=1}^{k-1} F\left(\frac{ni}{k}\right), \\
e_k &= \left| a_f - \left[ \frac{n(2k-1)}{2k} - \frac{n}{k} \sum_{i=1}^{k-1} F\left(\frac{ni}{k}\right) \right] \right| \tag{4} \\
&= \left| E[X] - \frac{n(2k-1)}{2k} + \frac{n}{k} \sum_{i=1}^{k-1} F\left(\frac{ni}{k}\right) \right|. \tag{5}
\end{aligned}
$$

Like for the floor model, $e_k$ requires only knowing $k$ statistics of $f$. The rounding model has an advantage over the floor model that there is no need to convert scores between different scales and perform normalization. One drawback is that it requires knowing $n$ (the expression for $m_i^k$ is dependent on $n$), while the floor model does not. In our experiments, we assume $n = 100$, but, in practice, it may not be clear what the agents' ground truth granularity is and may be easier to just deal with scores from 1 to $k$. Furthermore, it may seem unnatural to essentially ask people to rate items as "$\frac{100}{6}, 50, \frac{200}{6}$" rather than "$1, 2, 3$" (though the conversion between the score and $m_i^k$ could be done behind the scenes essentially circumventing the potential practical complication). One can generalize both the floor and rounding model by using a score of $s(n,k)_i$ for the $i$'th region. For the floor setting, we set $s(n,k)_i = i$, and for the rounding setting $s(n,k)_i = m_i^k = \frac{n(2i+1)}{2k}$.

## 4. Computational Simulations

The above analysis leads to the immediate question of whether the example for which $e_2 < e_3$ was a fluke or whether using fewer choices can actually reduce error under reasonable assumptions on the

generative model. We study this question using simulations with what we believe are the two most natural models. While we have studied the continuous setting where the full set of options is continuous over $(0, n)$ and the compressed set is discrete $\{0, \ldots, k-1\}$, we now consider the perhaps more realistic setting where the full set is the discrete set $\{0, \ldots, n-1\}$ and the compressed set is the same (though it should be noted that the two settings are likely quite similar qualitatively).

The first generative model we consider is a uniform model in which the values of the pmf for each of the $n$ possible values are chosen independently and uniformly at random. Formally, we construct a histogram of $n$ scores according to Algorithm 1. We then compress the full scores to a compressed distribution $p_k$ by applying Algorithm 2. The second is a Gaussian model in which the values are generated according to a normal distribution with specified mean $\mu$ and standard deviation $\sigma$ (values below 0 are set to 0 and above $n-1$ to $n-1$). This model also takes as a parameter a number of samples $s$ to use for generating the scores. The procedure is given by Algorithm 3. As for the uniform setting, Algorithm 2 is then used to compress the scores.

---

**Algorithm 1** Procedure for generating full scores in a uniform model

---

**Inputs**: Number of scores $n$

  scoreSum $\leftarrow 0$
  **for** $i = 0 : n$ **do**
    $r \leftarrow$ random(0,1)
    scores[$i$] $\leftarrow r$
    scoreSum $=$ scoreSum $+r$
  **for** $i = 0 : n$ **do**
    scores[$i$] $=$ scores[$i$] / scoreSum

---

**Algorithm 2** Procedure for compressing scores

---

**Inputs**: scores[], number of total scores $n$, desired number of compressed scores $k$

  $Z(n,k) \leftarrow \frac{n}{k}$                                                     $\triangleright$ Normalization
  **for** $i = 0 : n$ **do**
    scoresCompressed $\left[ \left\lfloor \frac{i}{Z(n,k)} \right\rfloor \right]$ += scores[$i$]

---

**Algorithm 3** Procedure for generating scores in a Gaussian model

---

**Inputs**: Number of scores $n$, number of samples $s$, mean $\mu$, standard deviation $\sigma$

  **for** $i = 0 : s$ **do**
    $r \leftarrow$ randomGaussian($\mu, \sigma$)
    **if** $r < 0$ **then**
      $r = 0$
    **else if** $r > n - 1$ **then**
      $r \leftarrow n - 1$
    ++scores[round($r$)]
  **for** $i = 0 : n$ **do**
    scores[$i$] = scores[$i$] / $s$

---

For our simulations, we used $n = 100$, and considered $k = 2, 3, 4, 5, 10$, which are popular and natural values. For the Gaussian model, we used $s = 1000$, $\mu = 50$, $\sigma = \frac{50}{3}$. For each set of simulations, we computed the errors for all considered values of $k$ for $m = 100,000$ "items" (each corresponding to a different distribution generated according to the specified model). The main quantities we are interested in computing are the number of times that each value of $k$ produces the lowest error over the $m$ items, and the average value of the errors over all items for each $k$ value.

In the first set of experiments, we compared performance between using $k$ = 2, 3, 4, 5, 10 to see for how many of the trials each value of $k$ produced the minimal error (Table 1). Not surprisingly, we see that the number of victories (number of times that the value of $k$ produced the minimal error) increases monotonically with the value of $k$, while the average error decreased monotonically (recall that we would have zero error if we set $k = 100$). However, what is perhaps surprising is that using a smaller number of compressed scores produced the optimal error in a far from negligible number of the trials. For the uniform model, using 10 scores minimized error only around 53% of the time, while using five scores minimized error 17% of the time, and even using two scores minimized it 5.6% of the time. The results were similar for the Gaussian model, though a bit more in favor of larger values of $k$, which is what we would expect because the Gaussian model is less likely to generate "fluke" distributions that could favor the smaller values.

**Table 1.** Number of times each value of $k$ in {2,3,4,5,10} produces minimal error and average error values, over 100,000 items generated according to both models.

|  | 2 | 3 | 4 | 5 | 10 |
|---|---|---|---|---|---|
| Uniform # victories | 5564 | 9265 | 14,870 | 16,974 | 53,327 |
| Uniform average error | 1.32 | 0.86 | 0.53 | 0.41 | 0.19 |
| Gaussian # victories | 3025 | 7336 | 14,435 | 17,800 | 57,404 |
| Gaussian average error | 1.14 | 0.59 | 0.30 | 0.22 | 0.10 |

We next explored the number of victories between just $k = 2$ and $k = 3$, with results in Table 2. Again, we observed that using a larger value of $k$ generally reduces error, as expected. However, we find it extremely surprising that using $k = 2$ produces a lower error 37% of the time. As before, the larger $k$ value performs relatively better in the Gaussian model. We also looked at results for the most extreme comparison, $k = 2$ vs. $k = 10$ (Table 3). Using two scores outperformed 10 8.3% of the time in the uniform setting, which was larger than we expected. In Figures 3 and 4, we present a distribution for which $k = 2$ particularly outperformed $k = 10$. The full distribution has mean 54.188, while the $k = 2$ compression has mean 0.548 (54.253 after normalization) and $k = 10$ has mean 5.009 (55.009 after normalization). The normalized errors between the means were 0.906 for $k = 10$ and 0.048 for $k = 2$, yielding a difference of 0.859 in favor of $k = 2$.

**Table 2.** Results for $k = 2$ vs. 3.

|  | 2 | 3 |
|---|---|---|
| Uniform number of victories | 36,805 | 63,195 |
| Uniform average error | 1.31 | 0.86 |
| Gaussian number of victories | 30,454 | 69,546 |
| Gaussian average error | 1.13 | 0.58 |

**Table 3.** Results for $k = 2$ vs. 10.

|  | 2 | 10 |
|---|---|---|
| Uniform number of victories | 8253 | 91,747 |
| Uniform average error | 1.32 | 0.19 |
| Gaussian number of victories | 4369 | 95,631 |
| Gaussian average error | 1.13 | 0.10 |

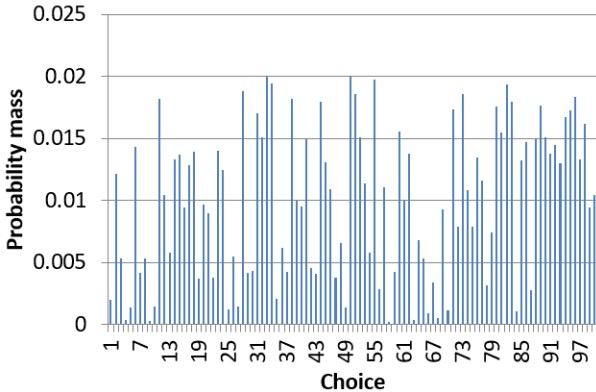

**Figure 3.** Example distribution where compressing with $k = 2$ produces significantly lower error than $k = 10$. The full distribution has mean 54.188, while the $k = 2$ compression has mean 0.548 (54.253 after normalization) and the $k = 10$ compression has mean 5.009 (55.009 after normalization). The normalized errors between the means were 0.906 for $k = 10$ and 0.048 for $k = 2$, yielding a difference of 0.859 in favor of $k = 2$.

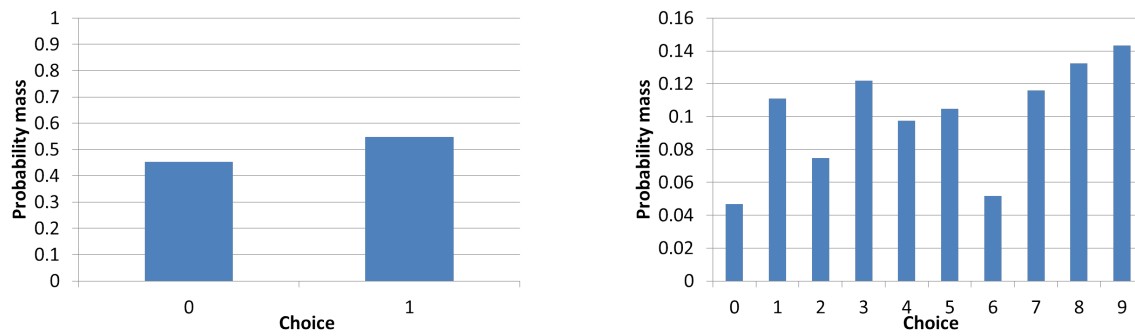

**Figure 4.** Compressed distribution for $k = 2$ vs. 10 for example from Figure 3.

We next repeated the extreme $k = 2$ vs. 10 comparison, but we imposed a restriction that the $k = 10$ option could not give a score below 3 or above 6 (Table 4). (If it selected a score below 3 then we set it to 3, and if above 6 we set it to 6). For some settings, for instance paper reviewing, extreme scores are very uncommon, and we strongly suspect that the vast majority of scores are in this middle range. Some possible explanations are that reviewers who give extreme scores may be required to put in additional work to justify their scores and are more likely to be involved in arguments with other reviewers (or with the authors in the rebuttal). Reviewers could also experience higher regret or embarrassment for being "wrong" and possibly off-base in the review by missing an important nuance. In this setting, using $k = 2$ outperforms $k = 10$ nearly $\frac{1}{3}$ of the time in the uniform model.

We also considered the situation where we restricted the $k = 10$ scores to fall between 3 and 7, as opposed to 3 and 6 (Table 5). Note that the possible scores range from 0–9, so this restriction is asymmetric in that the lowest three possible scores are eliminated while only the highest two are. This is motivated by the intuition that raters may be less inclined to give extremely low scores, which may hurt the feelings of an author (for the case of paper reviewing). In this setting, which is seemingly quite similar to the 3–6 setting, $k = 2$ produced lower error 93% of the time in the uniform model!

**Table 4.** Number of times each value of $k$ in {2,10} produces minimal error and average error values, over 100,000 items generated according to both models. For $k = 10$, we only permitted scores between 3 and 6 (inclusive). If a score was below 3, we set it to be 3, and above 6 to 6.

|  | 2 | 10 |
|---|---|---|
| Uniform number of victories | 32,250 | 67,750 |
| Uniform average error | 1.31 | 0.74 |
| Gaussian number of victories | 10,859 | 89,141 |
| Gaussian average error | 1.13 | 0.20 |

**Table 5.** Number of times each value of $k$ in {2,10} produces minimal error and average error values, over 100,000 items generated according to both generative models. For $k = 10$, we only permitted scores between 3 and 7 (inclusive). If a score was below 3, we set it to be 3, and above 7 to 7.

|  | 2 | 10 |
|---|---|---|
| Uniform number of victories | 93,226 | 6774 |
| Uniform average error | 1.31 | 0.74 |
| Gaussian number of victories | 54,459 | 45,541 |
| Gaussian average error | 1.13 | 1.09 |

We next repeated these experiments for the rounding compression function. There are several interesting observations from Table 6. In this setting, $k = 3$ is the clear choice, performing best in both models (by a large margin for the Gaussian model). The smaller values of $k$ perform significantly better with rounding than flooring (as indicated by lower errors) while the larger values perform significantly worse, and their errors seem to approach 0.5 for both models. Taking both compressions into account, the optimal overall approach would still be to use flooring with $k = 10$, which produced the smallest average errors of 0.19 and 0.1 in the two models (Table 1), while using $k = 3$ with rounding produced errors of 0.47 and 0.24 (Table 6). The 2 vs. 3 experiments produced very similar results for the two compressions (Table 7). The 2 vs. 10 results were quite different, with 2 performing better almost 40% of the time with rounding (Table 8), vs. less than 10% with flooring (Table 3). In the 2 vs. 10 truncated 3–6 experiments, 2 performed relatively better with rounding for both models (Table 9), and for 2 vs. 10 truncated 3–7, $k = 2$ performed better nearly all the time (Table 10).

**Table 6.** Number of times each value of $k$ produces minimal error and average error values, over 100,000 items generated according to both models with rounding compression.

|  | 2 | 3 | 4 | 5 | 10 |
|---|---|---|---|---|---|
| Uniform # victories | 15,766 | 33,175 | 21,386 | 19,995 | 9678 |
| Uniform average error | 0.78 | 0.47 | 0.55 | 0.52 | 0.50 |
| Gaussian # victories | 13,262 | 64,870 | 10,331 | 9689 | 1848 |
| Gaussian average error | 0.67 | 0.24 | 0.50 | 0.50 | 0.50 |

**Table 7.** $k = 2$ vs. 3 with rounding compression.

|  | 2 | 3 |
|---|---|---|
| Uniform number of victories | 33,585 | 66,415 |
| Uniform average error | 0.78 | 0.47 |
| Gaussian number of victories | 18,307 | 81,693 |
| Gaussian average error | 0.67 | 0.24 |

**Table 8.** $k = 2$ vs. 10 with rounding compression.

|  | **2** | **10** |
|---|---|---|
| Uniform number of victories | 37,225 | 62,775 |
| Uniform average error | 0.78 | 0.50 |
| Gaussian number of victories | 37,897 | 62,103 |
| Gaussian average error | 0.67 | 0.50 |

**Table 9.** $k = 2$ vs. 10 with rounding compression. For $k = 10$ only scores permitted between 3 and 6.

|  | **2** | **10** |
|---|---|---|
| Uniform number of victories | 55,676 | 44,324 |
| Uniform average error | 0.79 | 0.89 |
| Gaussian number of victories | 24,128 | 75,872 |
| Gaussian average error | 0.67 | 0.34 |

**Table 10.** $k = 2$ vs. 10 with rounding compression. For $k = 10$ only scores permitted between 3 and 7.

|  | **2** | **10** |
|---|---|---|
| Uniform number of victories | 99,586 | 414 |
| Uniform average error | 0.78 | 3.50 |
| Gaussian number of victories | 95,692 | 4308 |
| Gaussian average error | 0.67 | 1.45 |

## 5. Experiments

The empirical analysis of ranking-based datasets depends on the availability of large amounts of data depicting different types of real scenarios. For our experimental setup, we used two different datasets from the Preflib database [27]. One of these datasets contains 675,069 ratings on scale 1–5 of 1842 hotels from the TripAdvisor website. The other consists of 398 approval ballots and subjective ratings on a 20-point scale collected over 15 potential candidates for the 2002 French Presidential election. The rating was provided by students at Institut d'Etudes Politiques de Paris. The main quantities we are interested in computing are the number of times that each value of $k$ produces the lowest error over the items, and the average value of the errors over all items for each $k$ value. We also provide experimental results from the Jester Online Recommender System on joke ratings.

### 5.1. TripAdvisor Hotel Rating

In the first set of experiments, the dataset contains different types of ratings based on the price, quality of rooms, proximity of location, etc., as well as overall rating provided by the users scraped from TripAdvisor. We compared performance between using $k = 2, 3, 4, 5$ to see for how many of the trials each value of $k$ produced the minimal error using the floor approach (Tables 11 and 12). Surprisingly, we see that the number of victories sometimes decreases with the increase in value of $k$, while the average error decreased monotonically (recall that we would have zero error if we set $k$ to the actual maximum rating point). The number of victories increases for some cases with k = 2 vs. 3 compared to 2 vs. 4 (Table 13).

We next explored rounding to generate the ratings (Tables 14–17). For each value of $k$, all ratings provided by users were compressed with the computed $k$ midpoints and the average score was calculated. Table 14 shows the average error induced by the compression which performs better than the floor approach for this dataset. An interesting observation found for rounding is that using $k = n = 5$ was outperformed by using $k = 4$ for several ratings, using both the average error and number of victories metrics, as shown in Table 17.

**Table 11.** Average flooring error for hotel ratings.

| Average Error | $k = 2$ | 3 | 4 |
|---|---|---|---|
| Overall | 1.04 | 0.31 | 0.15 |
| Price | 1.07 | 0.27 | 0.14 |
| Rooms | 1.06 | 0.32 | 0.16 |
| Location | 1.47 | 0.42 | 0.16 |
| Cleanliness | 1.43 | 0.40 | 0.16 |
| Front Desk | 1.34 | 0.33 | 0.14 |
| Service | 1.24 | 0.32 | 0.14 |
| Business Service | 0.96 | 0.28 | 0.18 |

**Table 12.** Number of times each $k$ minimizes flooring error.

| Minimal Error | $k = 2$ | 3 | 4 |
|---|---|---|---|
| Overall | 235 | 450 | 1157 |
| Price | 181 | 518 | 1143 |
| Rooms | 254 | 406 | 1182 |
| Location | 111 | 231 | 1500 |
| Cleanliness | 122 | 302 | 1418 |
| Front Desk | 120 | 387 | 1335 |
| Service | 140 | 403 | 1299 |
| Business Service | 316 | 499 | 1027 |

**Table 13.** Number of times $k$ minimizes flooring error.

| # of Victories | $k = 2$ vs. 3 | 2 vs. 4 | 3 vs. 4 |
|---|---|---|---|
| Overall | 243, 1599 | 277, 1565 | 5, 1837 |
| Price | 187, 1655 | 211, 1631 | 4, 1838 |
| Rooms | 275, 1567 | 283, 1559 | 10, 1832 |
| Location | 126, 1716 | 122, 1720 | 11, 1831 |
| Cleanliness | 126, 1716 | 141, 1701 | 5, 1837 |
| Front Desk | 130, 1712 | 133, 1709 | 8, 1834 |
| Service | 153, 1689 | 152, 1690 | 11, 1831 |
| Business Service | 368, 1474 | 329, 1513 | 22, 1820 |

**Table 14.** Average error using rounding approach.

| Average Error | $k = 2$ | 3 | 4 |
|---|---|---|---|
| Overall | 0.50 | 0.28 | 0.15 |
| Price | 0.48 | 0.31 | 0.15 |
| Rooms | 0.48 | 0.30 | 0.16 |
| Location | 0.63 | 0.41 | 0.22 |
| Cleanliness | 0.6 | 0.4 | 0.21 |
| Front Desk | 0.55 | 0.39 | 0.21 |
| Service | 0.52 | 0.36 | 0.18 |
| Business Service | 0.39 | 0.36 | 0.18 |

**Table 15.** Number of times $k$ minimizes error with rounding.

| Minimal Error | $k = 2$ | 3 | 4 |
|---|---|---|---|
| Overall | 82 | 132 | 1628 |
| Price | 92 | 74 | 1676 |
| Rooms | 152 | 81 | 1609 |
| Location | 93 | 52 | 1697 |
| Cleanliness | 79 | 44 | 1719 |
| Front Desk | 89 | 50 | 1703 |
| Service | 102 | 29 | 1711 |
| Business Service | 246 | 123 | 1473 |

**Table 16.** Number of times $k$ minimizes error with rounding.

| # of Victories | $k = 2$ vs. 3 | 2 vs. 4 | 3 vs. 4 |
|---|---|---|---|
| Overall | 161, 1681 | 113, 1729 | 486, 1356 |
| Price | 270, 1572 | 101, 1741 | 385, 1457 |
| Rooms | 344, 1498 | 173, 1669 | 575, 1267 |
| Location | 275, 1567 | 109, 1733 | 344, 1498 |
| Cleanliness | 210, 1632 | 90, 1752 | 289, 1553 |
| Front Desk | 380, 1462 | 95, 1747 | 332, 1510 |
| Service | 358, 1484 | 109, 1733 | 399, 1443 |
| Business Service | 870, 972 | 278, 1564 | 853, 989 |

**Table 17.** # victories and average rounding error, $k$ in {4,5}.

| | | |
|---|---|---|
| Overall | Average error | 0.15, 0.21 |
| | # of victories | 1007, 835 |
| Price | Average error | 0.15, 0.17 |
| | # of victories | 955, 887 |
| Rooms | Average error | 0.15, 0.23 |
| | # of victories | 1076, 766 |
| Location | Average error | 0.22, 0.22 |
| | # of victories | 694, 1148 |
| Cleanliness | Average error | 0.21, 0.19 |
| | # of victories | 653, 1189 |
| Front Desk | Average error | 0.21, 0.17 |
| | # of victories | 662, 1180 |
| Service | Average error | 0.18, 0.18 |
| | # of victories | 827, 1015 |
| Business Service | Average error | 0.18, 0.31 |
| | # of victories | 1233, 609 |

### 5.2. French Presidential Election

We next experimented on data from the 2002 French Presidential Election (Tables 18 and 19). This dataset had both approval ballots and subjective ratings of the candidates by each voter. Voters rated the candidates on a scale of 20 where 0.0 is the lowest possible rating and −1.0 indicates a missing value (our experiments ignored the candidates with −1). The number of victories and minimal flooring error were consistent for all comparisons, with higher error achieved for lower $k$ values for each candidate.

On the other hand, with rounding compression, the minimal error was achieved for $k = 2$ for one candidate, while it was achieved for the two highest values $k = 8$ or 10 for the others.

**Table 18.** Average flooring error for French election.

| Average Error | 2 | 3 | 4 | 5 | 8 | 10 |
|---|---|---|---|---|---|---|
| Francois Bayrou | 3.18 | 1.5 | 0.94 | 0.66 | 0.3 | 0.2 |
| Olivier Besancenot | 1.7 | 0.8 | 0.5 | 0.35 | 0.16 | 0.1 |
| Christine Boutin | 1.15 | 0.54 | 0.34 | 0.24 | 0.11 | 0.07 |
| Jacques Cheminade | 0.64 | 0.3 | 0.19 | 0.13 | 0.06 | 0.04 |
| Jean-Pierre Chevenement | 3.69 | 1.74 | 1.09 | 0.77 | 0.35 | 0.23 |
| Jacques Chirac | 3.48 | 1.64 | 1.03 | 0.72 | 0.33 | 0.21 |
| Robert Hue | 2.39 | 1.12 | 0.7 | 0.49 | 0.22 | 0.14 |
| Lionel Jospin | 5.45 | 2.57 | 1.61 | 1.13 | 0.52 | 0.33 |
| Arlette Laguiller | 2.2 | 1.04 | 0.65 | 0.46 | 0.21 | 0.13 |
| Brice Lalonde | 1.53 | 0.72 | 0.45 | 0.32 | 0.14 | 0.09 |
| Corine Lepage | 2.24 | 1.06 | 0.67 | 0.47 | 0.22 | 0.14 |
| Jean-Marie Le Pen | 0.4 | 0.19 | 0.12 | 0.08 | 0.04 | 0.02 |
| Alain Madelin | 1.93 | 0.91 | 0.57 | 0.4 | 0.18 | 0.12 |
| Noel Mamere | 3.68 | 1.74 | 1.09 | 0.77 | 0.35 | 0.23 |
| Bruno Maigret | 0.31 | 0.15 | 0.09 | 0.06 | 0.03 | 0.02 |

**Table 19.** Average rounding error for French election.

| Average Error | 2 | 3 | 4 | 5 | 8 | 10 |
|---|---|---|---|---|---|---|
| Francois Bayrou | 1.65 | 0.73 | 0.91 | 0.75 | 0.48 | 0.62 |
| Olivier Besancenot | 3.88 | 2.39 | 2.14 | 1.7 | 1.31 | 1.25 |
| Christine Boutin | 3.87 | 2.39 | 1.84 | 1.5 | 0.9 | 0.86 |
| Jacques Cheminade | 4.34 | 2.72 | 2.07 | 1.65 | 1.02 | 0.88 |
| Jean-Pierre Chevenement | 1.47 | 0.65 | 1.2 | 0.82 | 0.55 | 0.61 |
| Jacques Chirac | 1.64 | 1.0 | 1.13 | 0.88 | 0.55 | 0.64 |
| Robert Hue | 2.51 | 1.27 | 1.14 | 1.09 | 0.67 | 0.77 |
| Lionel Jospin | 0.33 | 0.49 | 0.87 | 0.67 | 0.51 | 0.63 |
| Arlette Laguiller | 2.62 | 1.34 | 1.34 | 1.02 | 0.6 | 0.63 |
| Brice Lalonde | 3.45 | 1.9 | 1.55 | 1.21 | 0.66 | 0.78 |
| Corine Lepage | 2.89 | 1.59 | 1.56 | 1.16 | 0.79 | 0.87 |
| Jean-Marie Le Pen | 4.92 | 3.26 | 2.55 | 2.06 | 1.39 | 1.2 |
| Alain Madelin | 3.18 | 1.8 | 1.52 | 1.17 | 0.72 | 0.7 |
| Noel Mamere | 2.02 | 1.55 | 1.77 | 1.44 | 1.29 | 1.41 |
| Bruno Maigret | 4.88 | 3.23 | 2.46 | 1.99 | 1.28 | 1.1 |

*5.3. Joke Recommender System*

We also experimented on anonymous ratings data from the Jester Online Joke Recommender System [28]. Data was collected from 73,421 anonymous users between April 1999–May 2003 who have rated 36 or more jokes with ratings of real values ranging from −10.00 to +10.00. We included data from 24,983 users in our experiment. Each row of the dataset represents the rating from single user. The first column contains the number of jokes rated by a user and the next 100 columns give the ratings for jokes 1–100. Due to space limitations, we only experimented on a subset of columns (the ten most densely populated). The results are shown in Tables 20 and 21.

**Table 20.** Average flooring error for Jester dataset.

| Average Error | 2 | 3 | 4 | 5 | 10 |
|---|---|---|---|---|---|
| Joke 5 | 0.57 | 0.53 | 0.52 | 0.51 | 0.5 |
| Joke 7 | 1.32 | 0.88 | 0.74 | 0.66 | 0.54 |
| Joke 8 | 1.51 | 0.97 | 0.8 | 0.71 | 0.56 |
| Joke 13 | 2.52 | 1.45 | 1.09 | 0.91 | 0.61 |
| Joke 15 | 2.48 | 1.43 | 1.08 | 0.91 | 0.62 |
| Joke 16 | 3.72 | 2.01 | 1.44 | 1.16 | 0.69 |
| Joke 17 | 1.94 | 1.18 | 0.92 | 0.8 | 0.58 |
| Joke 18 | 1.51 | 0.97 | 0.79 | 0.71 | 0.56 |
| Joke 19 | 0.8 | 0.64 | 0.58 | 0.56 | 0.51 |
| Joke 20 | 1.77 | 1.1 | 0.87 | 0.76 | 0.57 |

**Table 21.** Average rounding error for Jester dataset.

| Average Error | 2 | 3 | 4 | 5 | 10 |
|---|---|---|---|---|---|
| Joke 5 | 0.48 | 0.47 | 0.48 | 0.47 | 0.48 |
| Joke 7 | 1.2 | 1.2 | 1.2 | 1.2 | 1.2 |
| Joke 8 | 1.44 | 1.43 | 1.42 | 1.43 | 1.42 |
| Joke 13 | 2.43 | 2.43 | 2.43 | 2.42 | 2.42 |
| Joke 15 | 2.34 | 2.34 | 2.33 | 2.33 | 2.33 |
| Joke 16 | 3.59 | 3.58 | 3.57 | 3.57 | 3.57 |
| Joke 17 | 1.84 | 1.82 | 1.82 | 1.81 | 1.81 |
| Joke 18 | 1.45 | 1.44 | 1.44 | 1.44 | 1.44 |
| Joke 19 | 0.72 | 0.72 | 0.71 | 0.71 | 0.71 |
| Joke 20 | 1.65 | 1.63 | 1.63 | 1.63 | 1.63 |

For the TripAdvisor and French election data, the errors decrease intuitively as the number of choices increase. However, surprisingly for the Jester dataset, we observe that the average errors are very close for all of the options ($k = 2, 3, 4, 5, 10$) with rounding compression (though with flooring they decrease monotonically with increasing $k$ value). These results suggest that, while using more options seems to generally be better on real data using our models and metrics, this is not always the case. In the future, we would like to explore deeper and understand what properties of the distribution and dataset determine when a smaller value of $k$ can outperform the larger ones.

## 6. Conclusions

Settings in which humans must rate items or entities from a small discrete set of options are ubiquitous. We have singled out several important applications—rating attractiveness for dating websites, assigning grades to students, and reviewing academic papers. The number of available options can vary considerably, even within different instantiations of the same application. For instance, we saw that three popular sites for attractiveness rating use completely different systems: Hot or Not uses a 1–10 system, OkCupid uses 1–5 "star" system, and Tinder uses a binary 1–2 "swipe" system. Despite the problem's importance, we have not seen it studied theoretically previously. Our goal is to select $k$ to minimize the average (normalized) error between the compressed average score and the ground truth average. We studied two natural models for generating the scores. The first is a uniform model where the scores are selected independently and uniformly at random, and the second is a Gaussian model where they are selected according to a more structured procedure that gives preference for the options near the center. We provided a closed-form solution for continuous distributions with arbitrary cdf. This allows us to characterize the relative performance of choices of $k$ for a given distribution. We saw that, counterintuitively, using a smaller value

of $k$ can actually produce lower error for some distributions (even though we know that, as $k$ approaches $n$, the error approaches 0): we presented specific distributions for which using $k = 2$ outperforms 3 and 10.

We performed numerous simulations comparing the performance between different values of $k$ for different generative models and metrics. The main metric was the absolute number of times for which values of $k$ produced the minimal error. We also considered the average error over all simulated items. Not surprisingly, we observed that performance generally improves monotonically with $k$ as expected, and more so for the Gaussian model than uniform. However, we observe that small $k$ values can be optimal a non-negligible amount of the time, which is perhaps counterintuitive. In fact, using $k = 2$ outperformed $k = 3, 4, 5$, and 10 on 5.6% of the trials in the uniform setting. Just comparing 2 vs. 3, $k = 2$ performed better around 37% of the time. Using $k = 2$ outperformed 10 8.3% of the time, and when we restricted $k = 10$ to only assign values between 3 and 7 inclusive, $k = 2$ actually produced lower error 93% of the time! This could correspond to a setting where raters are ashamed to assign extreme scores (particularly extreme low scores). We compared two natural compression rules—one based on the floor function and one based on rounding—and weighed the pros and cons of each. For smaller $k$ rounding leads to significantly lower error than flooring, with $k = 3$ the clear optimal choice, while for larger $k$ rounding leads to much larger error.

A future avenue is to extend our analysis to better understand specific distributions for which different $k$ values are optimal, while our simulations are in aggregate over many distributions. Application domains will have distributions with different properties, and improved understanding will allow us to determine which $k$ is optimal for the types of distributions we expect to encounter. This improved understanding can be coupled with further data exploration.

**Author Contributions:** Conceptualization, S.G.; data curation, F.B.Y.; formal analysis, S.G.; investigation, S.G. and F.B.Y.; methodology, S.G.; project administration, S.G.; writing–original draft preparation, S.G. and F.B.Y.; writing–review and editing, S.G.; visualization, S.G.; supervision, S.G.; project administration, S.G.

**Funding:** This research received no external funding.

**Conflicts of Interest:** The authors declare no conflict of interest.

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
