# Peer review of "Optimal Number of Choices in Rating Contexts"

_2504-2289, doi:10.3390/bdcc3030048_

Round 1

Reviewer 1 Report

The authors present an approach to evaluate the effect that the number of choices offered in contexts where users perform ratings. The authors' approach begins from the viewpoint that a "full-scale" rating domain exists, having available as many options as the number of items, providing thus the potential (but not the necessity) for a user to totally order the items according to the scores given.

On page 5, the authors exemplify their theoretical model using an example. However, it is not clear how this relates to a realistic example. Section 2 is in need of better explanations (especially the definitions of ak and ek) and exemplifications, linking the model to real-world cases.

On page 10, the authors state that they "strongly suspect" that in a peer review process extreme ratings, and especially extremely low ratings, are not common, attributing this behavior to some reservations on the reviewers' side. This should be better substantiated. Some statistics on reviews for a single conference are listed in https://nlpers.blogspot.com/2015/06/ The authors could use these data and additionally contact organizers of past conferences or journal offices to get some real-world data about the ratings of scientific papers.

The first example given on page 10 needs better explanation. The authors start off with a 10-grade rating scale, which they subsequently limit to use only 4 ratings. And later on, they map the (effective) four grade rating scale back to a 10-grade scale. Errors are bound to increase in this setting and be larger than a mapping to a 2-grade rating scale, since the distance between actual values and midpoints is also bound to increase. Furthermore, it is not clear what the argumentation stemming from this example is: if a four-grade rating scale was offered to reviewers, wouldn't they still avoid options "1" and "4"? Is a 2-grade rating scale appropriate for peer reviewing?

In the last paragraph of page 8, the correspondence between commentary and experiments should be made clear. For instance the text "produced the smallest average errors of 0.19 and 0.1" refers to the experiments summarized in table 3, however the reader should examine data values to make this mapping. Mapping references should be explicitly established in the text.

Minor comments:

On page 3, argminkek is not defined.

On page 4, pdf and cdf are not defined; while their meaning is fairly obvious, the acronym should be defined upon its first appearance. Furthermore, being an acronym, it should be spelled out in block letters.

On page 7, the region midpoints q1 and q2 are listed, however the way that midpoints are computed has not been described. A definition follows, but the definition should precede the example.

On page 8 the term "victories" is used, without having been defined. The term meaning becomes apparent as the text progresses, but it should be clear right from the first occurrence. 

Author Response

Section 2/Page 5 is entitled "theoretical characterization" and is presenting the theoretical model; realistic/real-world examples are discussed in other sections (e.g., introduction and experiment sections). For example, one realistic real-world situation where optimizing the number of choice options is important is online dating, where different sites use 2, 5, and 10 options.

To clarify, the "example" on page 5 is not a "real world example," it is just a specific concrete example of a distribution exemplifying the phenomenon we have described of 2 choices performing significantly better than 3.

a_k and e_k are defined at the bottom of page 2.

We can rephrase the comment on page 10 about suspecting the extreme ratings are uncommon to clarify that this is a pure hypothesis to be investigated further in future work and not a claimed scientific contribution of the current work.

Thank you for the link to the NAACL blog post; however while it presents several interesting graphs of analysis it does not show the distribution of the reviewers` scores, which is the quantity we are interested in. There is also an interesting blog post about a NIPS experiment, but again the data for score distributions is not provided http://blog.mrtz.org/2014/12/15/the-nips-experiment.html.

The primary goal of the current paper is to develop a new theoretical framework, showcased by our simulations. An interesting future paper would look more carefully at a specific example such as paper reviewing (we are not aware of any available data on the quantity we are interested in). This may involve running additional experiments (by talking to conference organizers and obtaining data) that are beyond the scope of this current project.

The point of the example in Table 4 (comparing k=2 vs. k=10 but only permitting 3-6) is as follows. We think that it is possible that certain real-world scenarios (e.g., those where reviewers or assessers are afraid to use extreme scores for a 1-10 scale) exist (possibly for paper reviewing, though data will need to be explored further in the future to confirm). Regardless, we think this is likely a natural phenomena and can occur in several real scenarios. We find it interesting, from a theoretical perspective, to see just how much this hurts performance and how often just using k=2 would have been better than k=10 with this restriction. Up until now no one knew, and we find it interesting to see that k=10 does better than k=2 around 2/3 of the time for the uniform case (though still around 90% for the Gaussian case). While we knew that errors were bound to increase, we did not know the exact extent of this increase.

Note that we are not prescribing what specific value of k should be used, nor making claims about the psychological behavior of reviewers.

My guess is that reviewers are less likely to utilize the "extreme" scores as the number of options k decreases. If a reviewer rates a paper a 1 or 10 on a 1-10 scale, then it will really stand out and they will likely have to put in more energy justifying their claim, while if there is a 1-2 or 1-4 scale, the extreme scores will be more common and not stand out as much. Of course experimental data is needed to explore this hypothesis.

I have seen conferences with 5-point scales (-2,-1,0,1,2). I see no reason why a 3 or even 2-point scale (1 for accept 0 for reject) might not be appropriate depending on the circumstances. Keep in mind that in computer science conferences the final decision is just accept or reject (there are not the further refined options of accepting with minor or major revisions that are common with journals).

I think the reviewer is referring to the final paragraph of page 10 (not of page 8 as stated) for the correspondence between the commentary and experiments, and we have added in further clarification to the text.

Minor comments:

Argmin is a standard mathematical term (https://en.wikipedia.org/wiki/Arg_max)

argmin_k e_k refers to the value of k that minimizes e_k

pdf and cdf are standard terms in mathematics/statistics. pdf is for probability density function and cdf is for cumulative distribution function. We have clarified this upon the first appearance.

The midpoint is the center of a region (the average of the left and right points). We work through the k=2 vs. k=3 example and present a general definition for arbitrary k.

We added in the meaning of "victories" when it is first used on page 8.

Reviewer 2 Report

The authors have face the problem of understanding when the use of different users’ ratings is really  optimal.  In particular,  users scores are modelled by means of  uniform random and Gaussian distributions. The authors analyze when using 2, 3, 4, 5, and 10 options out of a total of 100 is optimal in these models Some results are reported.

The authors propose a quite novel methodology to choose the optimal ranking scale, but presentation and technical qualities of the paper have to be improved.

For a possible publication, the paper lacks of a more detailed experimental section where the authors compare the proposed method with other techniques for ranking optimization, showing the real advantages with respect to the literature of the proposed methods especially for real applications (OSNs) . In addition, related work should be enriched with recent papers concerning methods based on different and more recent techniques based on data mining, artificial intelligence, game theory, cognitive computing and so on (that are the main topics of the journal).  Finally, I suggest a final linguistic revision of the entire paper to fix some typos and several unclear sentences.

Author Response

I am not sure what the terminology OSN in the review refers to.

The primary purpose of the paper is to present a new theoretical framework, with novel and interesting computational simulations. We are not claiming to have designed a new method that outperforms other methods experimentally, but rather a new theoretical and conceptual framework from which to evaluate the optimal choice of k to use for different applications.

We agree that a good direction for future study (for a subsequent publication) would be to compare the performance of several methods on a large dataset. In particular we think that the conference paper reviewing domain would be interesting for study (as mentioned in the other review and our response), and this may involve having to perform experiments and have discussions with conference organizers.

We have added in comparisons and references for several newer recent papers from data mining and artificial intelligence journals for related problems such as determining the optimal number of clusters to use for unsupervised learning algorithms.

We have performed spelling and grammar checks and revised several unclear sentences.

Round 2

Reviewer 1 Report

The authors have improved their paper to a degreee that warrants publication.

Reviewer 2 Report

The authors have addressed the requested reviews.